# SurvCNN: A Discrete Time-to-Event Cancer Survival Estimation Framework Using Image Representations of Omics Data

**DOI:** 10.3390/cancers13133106

**Published:** 2021-06-22

**Authors:** Yogesh Kalakoti, Shashank Yadav, Durai Sundar

**Affiliations:** 1DAILAB, Department of Biochemical Engineering & Biotechnology, Indian Institute of Technology (IIT) Delhi, New Delhi 110 016, India; yogesh.kalakoti@dbeb.iitd.ac.in (Y.K.); shashank.yadav1@alumni.iitd.ac.in (S.Y.); 2Indian Institute of Technology (IIT) Delhi, School of Artificial Intelligence, New Delhi 110 016, India

**Keywords:** multiomics, survival, lung adenocarcinoma, machine learning, CNN

## Abstract

**Simple Summary:**

Robust methods for modelling and estimation of cancer survival could be relevant in understanding and limiting the impact of cancer. This study was aimed at developing an efficient Machine learning (ML) pipeline that could model survival in Lung Adenocarcinoma (LUAD) patients. Image transformations of multi omics data were employed for training a machine vision-based model capable of segregating patients into high-risk and low-risk subgroups. The performance was evaluated using concordance index, Brier score, and other similar metrices. The proposed model was able to outperform similar methods with a high degree of confidence. Furthermore, critical modules in cell cycle and pathways were also identified.

**Abstract:**

The utility of multi-omics in personalized therapy and cancer survival analysis has been debated and demonstrated extensively in the recent past. Most of the current methods still suffer from data constraints such as high-dimensionality, unexplained interdependence, and subpar integration methods. Here, we propose SurvCNN, an alternative approach to process multi-omics data with robust computer vision architectures, to predict cancer prognosis for Lung Adenocarcinoma patients. Numerical multi-omics data were transformed into their image representations and fed into a Convolutional Neural network with a discrete-time model to predict survival probabilities. The framework also dichotomized patients into risk subgroups based on their survival probabilities over time. SurvCNN was evaluated on multiple performance metrics and outperformed existing methods with a high degree of confidence. Moreover, comprehensive insights into the relative performance of various combinations of omics datasets were probed. Critical biological processes, pathways and cell types identified from downstream processing of differentially expressed genes suggested that the framework could elucidate elements detrimental to a patient’s survival. Such integrative models with high predictive power would have a significant impact and utility in precision oncology.

## 1. Introduction

Molecular biology and high-throughput technology have significantly advanced in the past few decades, resulting in novel solutions for diagnostic and prognostic challenges in the way of personalized cancer therapy [1,2,3]. It has led to quantifying diverse omics-biomarkers in a clinically and economically feasible manner for an individual. This flexibility has allowed scientists to build robust personalized, predictive models using the generated data to evaluate prognostic dependencies such as clinical outcomes and probability of relapse [4,5,6,7]. Resources such as The Cancer Genome Atlas (TCGA), the International Cancer Genomics Consortium (ICGC), and the Cancer Cell Line Encyclopedia (CCLE), among others, have standardized the process of curating and hosting data from multiple studies, making them easily accessible [8,9,10]. These studies included cancers such as Lung Adenocarcinoma (LUAD) and squamous cell carcinoma (LUSC), which are known for their low survival rates and high chances of relapse [11]. A study suggested that most LUAD patients tend to be non-smokers, contrary to the general perception of a smoking-related basis to lung cancers [12].

Though multi-omics data can reveal significant insights into the mechanism of a disease like cancer, the form in which it is fed to a model is a decisive hyperparameter for its performance [13,14]. The good-old ‘Garbage-in, Garbage-out’ adage significantly reinforces itself when using multi-omics datasets [15]. While models try to resolve differential survival signatures from multi-omics data, understanding the relative importance of individual omics is equally important to improve cancer prognosis in a precision oncology setting. Initial attempts in this direction included Exploratory data analysis (EDA), where characterization and summarization of data are performed to detect any possible artifacts and outliers [16]. While latent variable methods in EDA aimed to identify significant contributors to variance globally, cluster analysis looked into pairwise Euclidean or any other metric to quantify delicate relationships within and among data [17]. Feature selection and extraction tend to be critical for the analysis to bear any concrete results owing to the high dimensionality of omics datasets. Principle component analysis (PCA), with its extensions including variable selection via L-1 penalized regularization, tends to be a go-to approach to mitigate the effects of such bottlenecks [18].

Apart from these conventional data-integration approaches, bright ideas like iCluster, CoxPath, and CNAmet have tried to mitigate the issue of high dimensionality and heterogeneity in omics to an extent, by consolidating the principles of mathematics and statistics in an effective manner [19,20,21,22]. However, the features derived from these methods are in the form of a feature vector (i.e., a vector of size p×n) and generally fed into machine learning algorithms with an assumption of mutual independence. In context with omics datasets, the local dependence of different features (genes) among themselves in such high dimensional feature space cannot be ignored. Machine learning setups work by learning the relationships among data that explain a given outcome, i.e., survival in our case. They are bound to perform poorly if the given data does not provide any information about the association among the covariates. Although modeling this dependence is not a naïve task, efforts like Seeded Bayesian Networks, Boolean Network models, and Copula methods have tried explaining these associations [23,24].

With a similar goal in mind, we intended to provide machine learning algorithms with some prior knowledge about the local interactions among the features (genes). Here, a simple yet efficient way to transform omics datasets into their image representations has been presented. Such images try to represent gene-associations, in addition to the numeric values already provided by omics datasets (gene expression, methylation, or miRNA). Further, these image representations of omics data types can be plugged in state of the art computer vision architecture such as Convolutional Neural Networks (CNN), which are tailor-made for learning such data and estimate the survival prognosis of a cancer patient [25]. The novelty of our method lies in the process of creating efficient representations of omics datasets, rather than being focused completely on the modelling strategy. Also, even though the datasets used are retrieved from public datasets, they have not been used in the proposed form for estimating survival, to the best of our knowledge. In summary, we have developed an ML-based workflow that is capable of estimating survival using image representations of omics data.

## 2. Materials and Methods

The underlying idea behind SurvCNN was to transform numeric omics data into an image form and feed it into a CNN with a custom loss function to predict survival probabilities for different time intervals. The calculated survival probabilities were used to fit a Kaplan–Meier curve for segregating patients in risk groups based on their conditional probabilities at different time intervals. The relative effectiveness of different combinations of omics data in estimating survival prognosis was also probed. The proposed workflow’s modular nature enabled the incorporation of multiple combinations of omics data types without drastically altering the underlying framework. For clarity, a detailed description of the twelve omics combinations analyzed is summarized in Table 1.

### 2.1. Datasets and Study Design

Semi-parametric proportional hazards regression (Cox-PH) is considered to be the most common strategy for generating survival estimates, where the relative risks of the patients (hazard ratios) are explained by the model covariates [26]. However, the model assumes a linear relationship between log-risk of failure and patient’s covariates, also referred to as the proportional hazard (PH) assumption. The PH assumption breaks quickly with high-dimensional multi-omics data with non-linear associations. Deep learning-based approaches mitigate this drawback due to their ability to effectively model non-linear relationships [27]. In tandem with deep learning models, Cox-PH models tend to perform well, given that the feature set is sufficiently small. Here an alternative approach has been employed that is described in the following sections.

### 2.2. Data Retrieval and Preprocessing

We employed three omics data types of LUAD, along with their clinical information in this study. All three data types, namely, mRNA-seq (illuminahiseq_rnaseqv2-RSEM_genes_normalized), methylation and miRNA-seq (illuminahiseq_mirnaseq-miR_gene_expression) were retrieved from firebrowse utility (http://firebrowse.org/ access date: 17 February 2020) of The Cancer Genome Atlas (TCGA) database. While firebrowse provided z-scaled RSEM values of mRNA expression, log2-RPM values for miRNA were retrieved and z-scaled. Common patients from the three datasets were extracted and the necessary processing (discard, impute) was done. While patients with more than 20 percent missing features were discarded, missing values (if any) in the rest were filled using R package impute [28]. In terms of the clinical dataset, TNM status (T: Tumor size and spread in tissue, N: spread in lymph nodes, M: metastasis) of the patients was included along with their age and gender. Survival data for the common patients were available in the form of Overall survival (OS) and clinical outcome (Binary). All data combinations and their general statistics are summarized in Table 2.

### 2.3. Feature Transformation

Omics datasets, which are inherently in numerical form, were transformed to an image for them to be compatible with CNN architectures. A general illustration is available in Figure 1A,B, where a feature vector corresponding to a datapoint is converted into a 2D feature matrix (Figure 1B). The location of every feature in the transformed image is dependent on the local relationships of genes in the original feature vector. In order to generate the 2D feature matrices, t-Distributed Stochastic Neighbor Embedding (t-SNE) and Uniform Manifold Approximation and Projection (UMAP) were employed [29,30]. These non-linear dimensionality reduction algorithms try to preserve the local relationships among features (genes) by calculating a similarity measure among features in high-dimensional space and optimizing the measures using a cost function.

Once the location of every feature was determined on the 2D feature matrix, grayscale intensities (0–255) corresponding to every gene/feature value in the original dataset (mRNA, methylation, and miRNA) were assigned to every element on the 2D feature matrix. For example, to build a survival prediction model using two omics data types and *n* patients, n×2 images were generated (two images per patient) using the proposed methodology. These images can be directly plugged into a deep learning architecture optimized for predicting survival probabilities. Similar attempts to transform non-image datasets into their image representations have been previously attempted [31]. Averaged image representation of the three omics data types for 446 patients are depicted in Figure 2. The effect of using different projection algorithms (t-SNE and UMAP) is evident from the differences in both sets.

Many hyperparameters such as perplexity, number of neighbors, distant metrics, and minimum distance were associated and needed to be optimized for these feature transformation methods. As an indirect measure of evaluating the goodness of projections, few genes involved in pathways reported to be dysregulated in cancer were mapped on the projections. t-SNE and UMAP plots mapped with 83 cancer-associated genes were analyzed with silhouette analysis and parameters that led to significant clustering were selected (Appendix A). The list of cancer-associated genes used to optimize the projections can be found in Appendix A. Also, additional explanation of the feature transformation protocol is described in Appendix A, Section 3, Appendix A (Feature transformation).

### 2.4. CNN Designs, Architecture, and Evaluation

Aiming for a modular framework, a parallel network architecture where different omics types can be simultaneously plugged into the model was constructed. This modularity allowed us to test for various combinations of omics datasets without altering the underlying architecture drastically. For each omics set the training and testing datasets were split in an 80:20 ratio. For example, the omics set I has 515 samples, so the training set and testing set contains 412 samples and 103 samples, respectively. Each convolution block contains one convolution layer with 256 nodes with a 3 × 3 kernel. It is followed by a batch normalization layer, ReLU activation layer, and a max-pooling layer with a 2 × 2 kernel. Batch normalization helps accelerate the training of large networks while max-pooling down-samples of the image at every layer [32]. The outputs of all the parallel layers were flattened, concatenated, and fed into a fully connected (FC) neural network Figure 1C. Additionally a detailed graphical illustration of a model architecture is depicted in Appendix A. As a measure to reduce overfitting, FC layers were regularized using dropout and early-stopping [33,34]. While dropout works by randomly turning off nodes from a given network layer with a given probability such that the model cannot rely upon the training data entirely and overfit, early-stopping halts the model training as soon as the generalization error starts increasing [33,34]. Pictorial representations of all the model architectures are compiled in Appendix A.

The proposed model provides flexibility in opting for the proportional hazards assumption versus its violation. For instance, with the proportional hazard assumption, the final dense layer was setup as a single neuron with no bias. Then it was connected to the n-dimensional model output where ‘n’ is the number of time intervals. On the other hand, in the case of large datasets where the proportional hazard assumption may get violated, the single node proportional hazard layer is skipped and the output of the fully connected dense layers is directly connected to an n-dimensional output layer. Every model has an n-dimensional output where n is the number of time intervals, in which total survival time (5 years, 10 years) is split. We opted for a bin width of 3 months for a 10-year survival analysis that gave us a 39-dimensional output layer.

### 2.5. Survival Analysis

The Cox-PH model is usually incorporated into deep neural networks as a final layer to give survival estimates [35,36,37]. However, as previously discussed, the assumption of proportionality is doubtful when the number of covariates is large. Also, stochastic gradient descent requires a single data point per epoch to run efficiently, whereas due to the PH assumption, the model output of all the data points having a higher survival duration than the given data point needs to be included for the calculation. These factors ultimately lead to memory bottlenecks, delayed convergence, and the possibility of loss function not reaching its global minima.

To mitigate these issues, a discrete-time survival method for modeling omics data and estimation of survival prognosis was employed. For the approach to work, a custom loss function was based on the negative partial likelihood, assuming the PH criteria had to be defined (Equation (1)). It was critical as major deep learning frameworks like TensorFlow did not have the required loss function built in. The discrete-time survival models are flexible as they do not assume the validity of the PH assumption. These models could easily be trained with SGD, which was attractive as this enabled rapid experimentation for optimizing the model hyperparameters. In discrete-time survival models, the total time frame (e.g., one year/five years/ten years) was divided into a set of intervals with a fixed duration (e.g., three months/six months). For each interval, the conditional probability of survival was estimated, given that an individual has survived up to the beginning of an interval. It was later used to classify the given patient into a high/low-risk class.

### 2.6. Loss Function

Negative log-likelihood was adapted as a loss function for the survival model [38]. For each time interval j, the neural network loss function was defined as:(1)$$loss=-\left(\sum_{i=1}^{d_{j}}\ln\left(1-h_{j}^{i}\right)+\sum_{d_{j+1}}^{r_{j}}\ln\left(h_{j}^{i}\right)\right)$$
where $h_{j}^{i}$ was the hazard probability for individual i for the $j^{th}$ time interval, there were $r_{j}$ individuals in observation during the $j^{th}$ time interval (these individuals had survived before the $j^{th}$ time interval) and the first $d_{j}$ of them suffered a failure during this interval. Ideally, the model should assign a higher and lower hazard value for failure and survival events, respectively. The defined loss function makes sure that the objective function is penalized in the event of the model giving low hazard for failure and high hazard for survival. The sum of each time interval’s losses gave us the overall loss, which was propagated back to the network such that the model weights were iteratively altered to reduce the overall loss after each iteration and hence, ‘train’ itself. For the observed data, total log-likelihood is the sum of the log-likelihoods for every individual. Formally stating, for the neural network-based survival prediction model’s training, the goal was to maximize the likelihood; hence, minimization of the negative log-likelihood was performed using the SGD algorithm.

### 2.7. Hazard Probability

The proposed survival prediction model, with the custom loss function, naturally integrated non-proportional hazards and time-varying baseline hazard rates. The baseline hazard probability is allowed to vary freely with a time interval, but the effect of input data on hazard rate does not vary with follow-up time. In other words, if a specific combination of input data results in a high rate of death in the early follow-up period, it will also result in a high rate of death in the late follow-up period. This was implemented by setting the final hidden layer to have a single neuron and densely connecting the prior hidden layer to the final hidden layer without any bias weights. The neural network gave a separate hazard rate for each time interval J as output and had dimension t x 1,  where *t* was the number of time intervals. Given a set of time intervals, SurvCNN predicted the conditional probability of survival at every interval, i.e., $P(T>t_{j}|T>t_{j-1})$ for j=1, 2, …,J. The marginal probability (effective probability of survival at a particular time point) was also calculated by the product of all the conditional probabilities up to the $j^{th}$ time interval:(2)$$P\left(T>t_{j}\right)=\prod_{k=1}^{j}P(T>t_{k}|T>t_{k-1})$$

### 2.8. Dichotomizing Patient Groups Using Kaplan Meier Estimates

The Kaplan Meier (KM) survival curve estimated the survival rates and hazard for a given time interval (Figure 1D). The survival rate for any given time intervals was calculated as follows.
(3)$$S_{t}=\frac{number\;of\;individuals\;survived\;longer\;than\;t}{total\;number\;of\;individuals\;under\;study}$$

The KM survival curves were generated by segregating patients into two risk groups based on the median survival probability after a definite amount of time (one year, five years, or ten years). The two survival populations were statistically compared by testing the null hypotheses and calculating the associated *p*-values. In survival statistics, the log-rank *p*-value was calculated to estimate the statistical significance among the segregated risk groups [39].

### 2.9. Performance Metrics

Three performance metrics were employed to quantify the goodness of the proposed survival models. The first metric used was Harrel’s concordance index (C-index) that evaluates the relative ordering of events in time-series data [40]. C-index ranges between 0 and 1 (higher the better). It is a generalized version of area under the ROC curve and in survival analysis a higher C-index would suggest that the model’s ability to distinguish between high-risk and low-risk subgroups. A model with c-index > 0.7 indicates a good model performance. It has been extensively used in the literature to evaluate the effectiveness of survival prognosis models [36,37]. Lifelines python package was employed for implementing C-index as a performance metric. Secondly, the Brier score, which calculates the mean squared error between calculated survival probabilities and the actual survival status at every time point, was used with the help of scikit-learn package [41,42]. It ranges between 0 and 1 (lower the better). However, other methods have reported a 1-Brier score instead, hence in our case the Brier values should be read as higher the better. Additionally, the Inverse Probability of Censoring Weights (C-IPCW) was also employed to evaluate the models’ performance [43]. This metric takes into account the effect of censoring on the model estimates. It reweights the individuals that do not drop out by the inverse of their probability of not dropping out given covariates. Its range lies between 0 and 1 (higher the better.)

### 2.10. Functional Enrichment and Gene Ontology Analysis for Identified Biomarkers

Differentially expressed gene clusters among the two classes of patients were identified and exported for functional analysis. Enrichr, a web-based open-source enrichment toolbox, was employed for the analysis [44,45]. Statistically significant enrichments in KEGG 2019 Human pathways, GO Biological processes (2018), and Cell types (Human gene atlas) for the aberrated gene sets in the two prognostic subclasses were targeted [46,47,48]. The gene list for each of the classes is provided in Appendix A.

## 3. Results

To generate robust representations of omics datasets with predictive ability, we proposed a workflow that transforms numerical omics datasets into an image form while retaining critical information about feature dependencies. Multiple CNNs with different omics combinations were trained to predict survival statistics of LUAD cancer patients effectively. Irrespective of the omics data types included, the performance of all the experiments was evaluated by three metrics C-index, Brier score, and IPCW. To minimize the chances of over-fitting, five-fold cross-validation was performed for every run. Ultimately, differentially expressed genes were derived among the two risk groups and used for further analysis (Figure 3A).

### 3.1. Image Representations of Omics-Data Have Predictive Ability

Contrary to the conventional practice of feeding numerical omics data into machine learning frameworks, the independent values were converted into well-organized images. These images performed exceedingly well in predicting the patient-specific survival probabilities with a high degree of concordance. The complete set of performance metrics evaluating twelve omics-combinations each for two methods (t-SNE and UMAP) are compiled in Appendix A. With these results in hand, it can be hypothesized that such representations effectively culminate the local interactions among genetic features, which assists the machine learning algorithm in learning these relationships. This worked as it is much easier for the ML algorithm to capture and learn the associations in omics data if some prior information is provided. As an added bonus, the method also reduced the amount of hardware requirements to store the omics datasets by almost 90%.

### 3.2. Identification of Prognostic Subtypes in Lung Adenocarcinoma

To stratify Lung adenocarcinoma patients into two subclasses, the conditional probability of survival was calculated for each individual at every time point. Kaplan–Meier estimates were employed to segregate patients into high-risk (C1) and low-risk (C2) subgroups based on these probabilities. The two risk groups were stringently checked for significance and any experiment with log-rank *p*-value < 0.005 was not considered for the results. Though most omics combinations did well to segregate prognostic subtypes significantly, a combination of mRNA-seq and methylation performed exceedingly well (Figure 3B,C). Moreover, both internal and external validation further reinforced the robustness of the models as the two subclasses indeed had stark outcomes (Appendix A).

### 3.3. Integrating Omics-Types Affects the Performance of Prognostic Models

Among all the multi-omics combinations tested, an upward trend for mean-concordance values was observed as more omics-types were included. However, for the instances where miRNA-seq data were included, the performance dipped significantly (Table 3). The exact reason for this behavior can be attributed to the lower-dimensional character of miRNA-seq data. However, it is evident from the results that other multi-omics combinations outperformed single-omics models consistently. To verify if the performance of the tested omics combinations is indeed significantly different, we performed a pairwise t-test for C-indices. Looking at the t-statistics (Table 3), it can be safely eluded that the combination of mRNA-seq and methylation data, along with their clinical information, serves as the best predictor of survival than most of the combinations tested.

### 3.4. Transformation Based Multi-Omics Integration Outperforms Alternative Approaches

Our approach was compared with existing methods such as Cox-PH, RF-S, and Cox-nnet [26,36,49]. For each omics dataset (or datasets), the model was trained on 80% of randomly selected samples. The remaining 20% of the data (holdout set) was utilized to evaluate the model’s performance. Again, it is emphasized that each run of the model was replicated five times (five-fold cross-validation) to minimize the chances of generating an overfitted model. Also, no datapoint from the holdout set is ever shown to the training algorithm.

The comparison of performance in terms of the C-index of omics data among the four methods over twelve omics datasets is shown in Figure 4. SurvCNN was able to outperform other methods in nine out of the twelve omics sets tested. The combination of mRNA and methylation data, with (omics set X) and without (omics set IV) clinical information, proved to be the best predictor of survival. It must be noted that while SurvCNN performed relatively well than the competition, it lagged on instances where the primary dataset was miRNA-seq. This might be attributed to the low dimensionality of the miRNA-seq dataset, which might not have been captured efficiently by the feature transformation algorithm. In addition to Cox-PH and RF-S, our results were also compared with Cox- nnet, wherein they analyzed the prognosis prediction performance among omics-data sets using similar metrics as ours.

While our best model (including mRNA-seq, methylation and clinical) registered a C-IPCW in the range of 0.68–0.73, Cox-nnet stayed in the 0.58–0.65 range throughout the omics combinations tested. This was in line with a previous work wherein it was around 0.6 (range: 0.55–0.59) for the LUAD dataset [36]. Similar trends were observed for the Brier score (Cox-nnet: 0.81–0.83; SurvCNN: 0.84–0.85) [36].

It should be noted that while C-indices for Cox-PH, RF-S and Cox-nnet were computed on LUAD data in-house, the Brier score (as 1-Brier score for compatibility with other performance metrics) for Cox-nnet was adapted from its original publication as both used the same (LUAD) dataset. Moreover, to verify if the performance improvement depicted by SurvCNN is statistically significant, a paired t-test was performed on every omics combination. As tabulated in Table 4, it can be asserted that SurvCNN outperforms the competition with a high degree of confidence in the majority of omics combinations. It only lags in two sets where miRNA-seq data is in abundance for the probable reasons mentioned earlier. Further, the choice of feature representation did not affect the performance of the models significantly. As evident from Figure 5A, selecting t-SNE over UMAP or vice-versa did not alter the outcome marginally. However, t-SNE was reported to be marginally better in terms of the Brier score.

### 3.5. SurvCNN Identifies LUAD Associated Pathways, GO Terms, and Cell Types

Differential expression levels among the two prognostic subtypes were quantified by averaging out the images corresponding to the two subclasses and filtering out the common pixel locations. As depicted in Figure 3A, genes associated with differential pixel intensities in the first and fourth quartile were selected as enrichment candidates for further analysis and 236 downregulated and 173 upregulated genes from images corresponding to the mRNA-seq data were obtained by the described method. Figure 6 shows the top ten significantly (*p*-value < 0.005, odds-ratio > 4.0) enriched terms associated with Pathway, Biological process, and cell type. More cell cycle and DNA replication-associated genes like CDK1, MCM4, CHEK1, and RFC5 were upregulated in C1 than in C2 [50,51,52,53]. We also identified genes corresponding to CD105^+^, CD71^+^, and CD34 endothelial such as RAD51, KIAA0101, PRC1, POLE2, CDK2, CDK1, CKS2, CENPN, and TAF5. Interestingly, anti-angiogenetic treatment targeting such tumor endothelial cells provides a survival advantage in the treatment of NSCLC [54]. On the other hand, the downregulated genes were mapped to Purine metabolism pathways, Oxytocin signaling pathway, among others (Appendix A). Also, genes associated with lung were mapped while enriching cell types (Human Gene Atlas) for the downregulated genes.

## 4. Discussion

The proposed SurvCNN method produced promising results and cemented the utility of multi-omics data in the lung cancer survival prognosis. Unlike other algorithms, it integrates Cox-PH regression and CNN models seamlessly into a single package. It not only makes the pipeline robust but also adds a modular aspect to it. This modularity has been exploited to test for various combinations of multi-omics datasets without drastically altering the underline architecture. Although most of the omics combinations tested outperformed the competition, the combination of mRNA-seq, miRNA-seq, and their clinical features proved to be the most effective omics-combination among the twelve sets tested. It also reinforced the utility of multi-omics in cancer research instead of a more traditional single-omics approach. Though only PH models were considered for the final comparison, model performances for PH and non-PH models are summarized in Appendix A.

It should be noted that creating image representations of omics-datasets was one of the fundamental requirements for this approach to work. The effectiveness of these representations is reflected in above-average prognosis results quantified by the C-index and log-rank test. It should be emphasized that all the performance metrics results are reported for the test-set only. The detailed record of all the training and testing results, along with comparative analysis, are compiled in Appendix A (Sheet4–Sheet7). Apart from identifying related pathways, the enrichment analysis also revealed critical biological functions and associated cell-types. DNA replication (GO:0006260), DNA metabolic processes (GO:0006259), DNA-dependent DNA replication (GO:0006261), mitotic cell cycle phase transition (GO:0044772), and G1/S transition of the mitotic cell cycle (GO:0000082) were among the top-ten associated functions (*p*-value < 0.005, odds-ratio > 6) for the overexpressed genes in C1. On the other hand, regulation of histone acetylation (GO:0090240), chemical homeostasis within a tissue (GO:0048875) and amino-acid transport (GO:0089718) were found to be downregulated in C1. These observations underline the cancerous characteristics that differentiate C1 from C2. It also reinforces the criticality of these processes that lead to aggressive cancer metastasis, tumor development, and possibly the ultimate fate of an individual [55].

Though most of the multi-omics combinations tested in the study were found to have above-average predictive ability, a few combinations did not perform as expected (Figure 5A,B. The principal component of these under-performing sets was miRNA-seq data. Contrasting feature dimensions (mRNA-seq:miRNA-seq: 500:1) and incompatibility with image transformations could be the two reasons for the same. It would be safe to say that very high dimensional datasets such as mRNA-seq and methylation may gain from our suggested methodology while low dimensional datasets are better off in their original numerical state. However, refinement of our transformation algorithm might also improve the predictability of the miRNA-seq dataset. In totality, it can be confidently asserted that image representations of omics data not only have a better survival prognosis predictive ability but also simplify the various aspects of managing omics datasets.

## 5. Conclusions

This work successfully demonstrated that numerical multi-omics data, transformed into their image representations, could extract meaningful information about the individual’s genomic profile. These representations of genomic information also simplify the process of identifying genetic clusters that are coregulated in a diseased individual and predict its likelihood of survival in terms of hazard ratios. SurvCNN also fared well against other machine learning-assisted survival prognosis models. Though we have primarily focused on its application on cancer prognosis detection, one can extrapolate the algorithms for use in a wide variety of applications.

As an immediate follow-up, given the encouraging results with LUAD datasets, we propose to scale this method into a holistic cancer prognosis prediction system in a future study. It would include data for a variety of cancer subtypes as opposed to the current single cancer multi-omics. The system would provide prognostic predictions to cancer patients based on their genomic profile and disease history. With more sophisticated CNN architectures and powerful hardware, our method and its derivatives can harness the full potential of recent advances in computer vision and prove to be a stepping-stone in the development of personalized therapy.

## Figures and Tables

**Figure 1 cancers-13-03106-f001:**
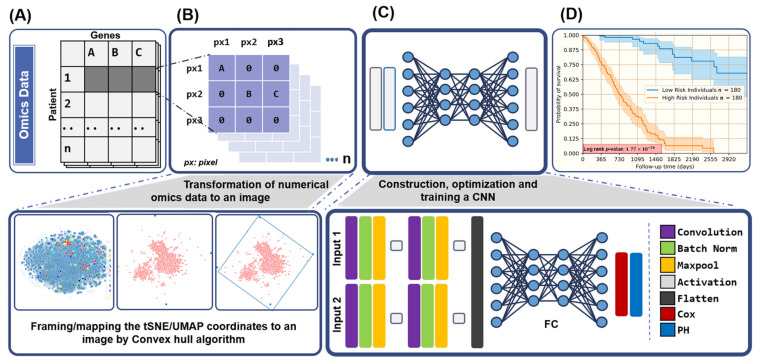
Schematic depicting the general workflow for SurvCNN. (**A**) Multi-omics data retrieval, (**B**) feature transformation, (**C**) building, training, and optimizing a deep neural network and (**D**) downstream analysis and inference.

**Figure 2 cancers-13-03106-f002:**
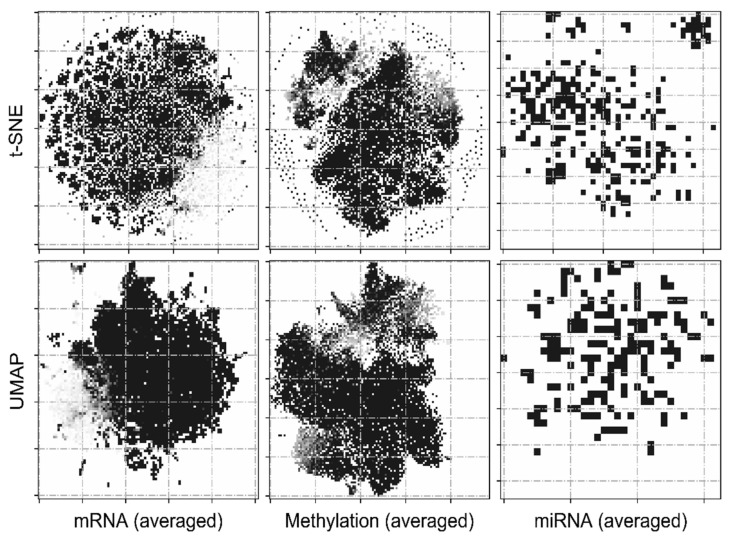
Graphical representations of omics datasets generated via t-SNE and UMAP. Every pixel in the image can be attributed to a gene, while its intensity quantifies the corresponding expression levels.

**Figure 3 cancers-13-03106-f003:**
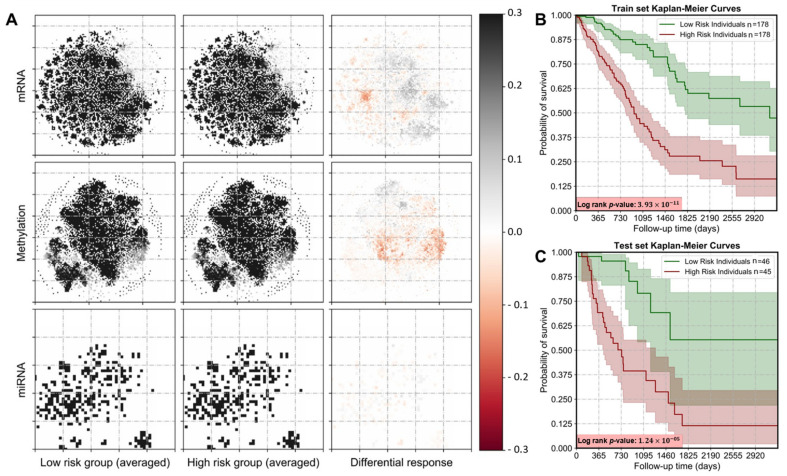
Differential analysis and survival classes. (**A**) Image representations for all the omics types for the two patient risk groups depicting the differential genes (marked with colors). Kaplan–Meier plots for (**B**) train and (**C**) test sets showing that SurvCNN can segregate patients into two risk groups with a high degree of confidence (log-rank *p*-value).

**Figure 4 cancers-13-03106-f004:**
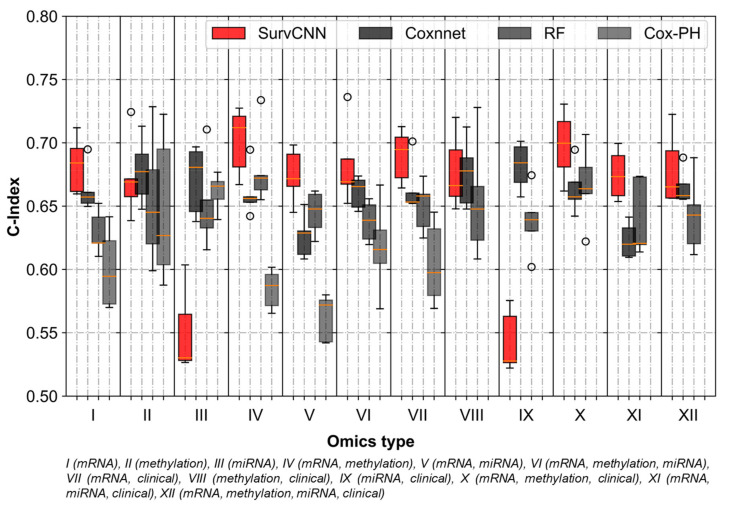
Comparison of performance measure (C-index) with existing methods. The effectiveness of SurvCNN’s novel feature transformation approach is highlighted with the fact that it outperforms the competition in nine out of twelve omics sets tested.

**Figure 5 cancers-13-03106-f005:**
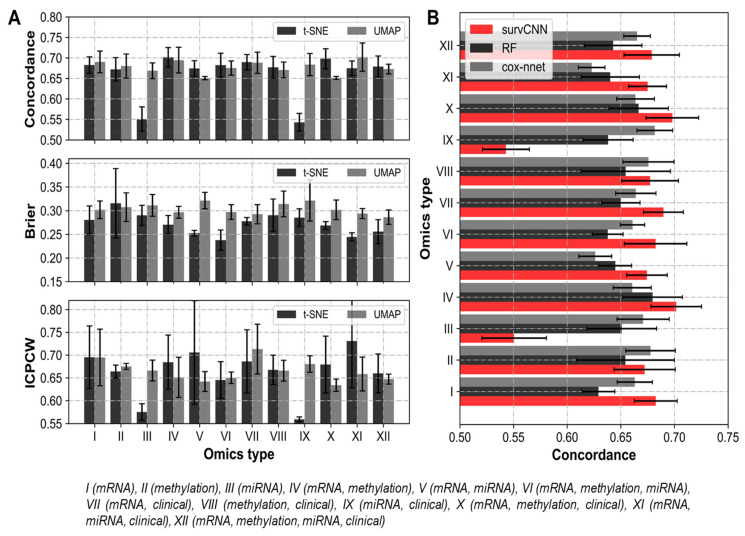
Overall performance of proposed workflow on different metrics of performance and projection algorithm. (**A**) Performance in terms of C-Index, Brier score and ICPCW (**B**) SurvCNN outperforms competing methods on nine out of twelve omics combinations tested.

**Figure 6 cancers-13-03106-f006:**
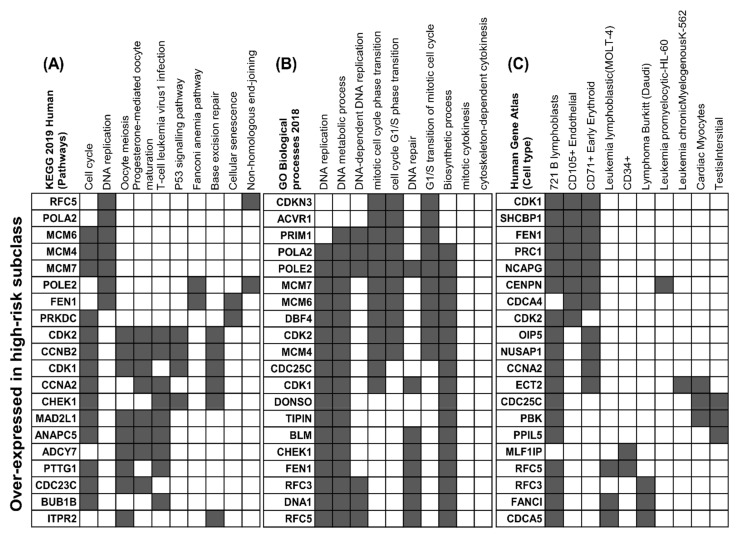
Enriched terms from KEGG 2019 Human pathways, GO Biological processes, and Cell type (Human gene atlas) for (**A**–**C**) overexpressed gene-set.

**Table 1 cancers-13-03106-t001:** Reference table for the twelve combinations of omics data analyzed in the study. For every omics set denoted by roman numerals, the ‘+’ sign implies the inclusion of the corresponding omics data type.

	I	II	III	IV	V	VI	VII	VIII	IX	X	XI	XII
***mRNA***	+			+	+	+	+			+	+	+
***meth***		+		+		+		+		+		+
***miRNA***			+		+	+			+		+	+
***Clinical***							+	+	+	+	+	+

**Table 2 cancers-13-03106-t002:** Overall statistics for all the omics data-type combinations for SurvCNN.

Omics Type *	Total Cases	Living	Deceased	Features Before Processing	Features After Processing	Age (yrs.)		Survival (yrs.)	
						*Median*	*Range*	*Median*	*Range*
**I**	515	328	187	20,172	123 × 123	66.0	38–88	1.803	0–2
**II**	458	293	165	17,052	123 × 123			1.786	
**III**	450	286	164	477	42 × 42			1.789	
**IV**	454	290	164	37,224	(123 × 123) × 2			1.785	
**V**	446	283	163	20,649	(123 × 123) × 2			1.788	
**VI**	446	283	163	37,701	(123 × 123) + (42 × 42)			1.788	
**VII**	515	328	187	20,192	(123 × 123) + 20			1.802	
**VIII**	458	293	165	17,072	(123 × 123) + 20			1.786	
**IX**	450	286	164	497	(42 × 42) + 20			1.789	
**X**	454	290	164	37,244	(123 × 123) × 2 + 20			1.784	
**XI**	446	283	163	20,669	(123 × 123) + (42 × 42) + 20			1.787	
**XII**	446	283	163	37,721	(123 × 123) × 2 + (42 × 42) + 20			1.787	

*Note: I (mRNA), II (methylation), III (miRNA), IV (mRNA, methylation), V (mRNA, miRNA), VI (mRNA, methylation, miRNA), VII (mRNA, clinical), VIII (methylation, clinical), IX (miRNA, clinical), X (mRNA, methylation, clinical), XI (mRNA, miRNA, clinical), XII (mRNA, methylation, miRNA, clinical).

**Table 3 cancers-13-03106-t003:** Performances comparison with different combinations of multi-omics data by pairwise paired t-test, according to C-index among five-fold cross-validation results. Note: Negative t-statistic indicates that Set1 is better than Set2. *p*-value < 0.05.

		Omics Type * (Set 1)										
		II	III	IV	V	VI	VII	VIII	IX	X	XI	XII
**Omics type * (Set 2)**	**I**	1.56	18.23	−5.21	2.32	0.021	−3.73	1.18	32.72	**−4.50**	4.50	0.80
	**II**	-	20.99	−3.64	−0.28	−3.78	−2.25	−1.00	17.19	**−4.09**	−0.36	−1.20
	**III**	-	-	−15.78	−14.89	−38.86	−15.70	−36.67	1.49	**−19.84**	−16.03	−49.08
	**IV**	-	-	-	6.50	2.15	4.29	3.90	23.57	**1.40**	6.33	3.13
	**V**	-	-	-	-	−1.05	−5.56	−0.56	30.39	**−7.21**	−0.23	−0.73
	**VI**	-	-	-	-	-	0.87	1.34	22.37	**−2.31**	0.97	0.91
	**VII**	-	-	-	-	-	-	2.14	28.09	**−2.42**	8.36	1.72
	**VIII**	-	-	-	-	-	-	-	40.89	**−5.17**	0.46	−0.84
	**IX**	-	-	-	-	-	-	-	-	**−30.35**	−35.70	−43.84
	**X**	-	-	-	-	-	-	-	-	-	**5.88**	**3.55**
	**XI**	-	-	-	-	-	-	-	-	-	-	−0.74

*Note 1: t-test statistics of the best omics combination are indicated in bold. Note 2: I (mRNA), II (methylation), III (miRNA), IV (mRNA, methylation), V (mRNA, miRNA), VI (mRNA, methylation, miRNA), VII (mRNA, clinical), VIII (methylation, clinical), IX (miRNA, clinical), X (mRNA, methylation, clinical), XI (mRNA, miRNA, clinical), XII (mRNA, methylation, miRNA, clinical).

**Table 4 cancers-13-03106-t004:** Pairwise t-test to validate the statistical significance of improved performance by SurvCNN. A positive test metric (t) denotes the superiority of SurvCNN. Annotations in bold mark the instances where SurvCNN significantly (*p* < 0.005) outperforms the competition.

	Omics Type *											
	I	II	III	IV	V	VI	VII	VIII	IX	X	XI	XII
**RF**	**13.52**	1.78	−22.69	2.29	**11.36**	4.84	**28.40**	2.71	−15.36	**8.46**	**5.66**	**9.16**
**Cox-PH**	**18.81**	1.63	−10.23	**23.75**	**25.03**	**12.15**	**13.61**	-	-	-	-	-
**Cox-nnet**	3.96	−1.10	−12.68	5.23	**11.59**	2.02	3.64	0.44	−26.87	**6.61**	**19.10**	1.97

*Note 1: Statistically significant values (*p* < 0.005) are marked in bold. Note 2: I (mRNA), II (methylation), III (miRNA), IV (mRNA, methylation), V (mRNA, miRNA), VI (mRNA, methylation, miRNA), VII (mRNA, clinical), VIII (methylation, clinical), IX (miRNA, clinical), X (mRNA, methylation, clinical), XI (mRNA, miRNA, clinical), XII (mRNA, methylation, miRNA, clinical).

## Data Availability

All the codes and data used for the study can be accessed at https://github.com/TeamSundar/SurvCNN.

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
