# Peer review of "SurvCNN: A Discrete Time-to-Event Cancer Survival Estimation Framework Using Image Representations of Omics Data"

_cancers, 2021, doi:10.3390/cancers13133106_

Round 1

Reviewer 1 Report

SurvCNN: Redefining the Utility of Multi-Omics Data in Cancer Survival Analysis

The authors presented a multiomic method for developing an efficient Machine learning (ML) pipeline that could model survival in Lung Adenocarcinoma (LUAD) patients. They claimed that they propose SurvCNN, to predict cancer prognosis for LUAD patients.

The article is well-written and interesting. I personally like this study.

Major:

  • This paper used some interesting terms which indicates that despite the too big claim, the whole study paper’s novelty might summarized in using SurvCNN for LUAD patients. What is your novelty here? SurvCNN is already proposed, and your datasets are also publicly available, please clarify this in the paper.
  • Authors claimed that: “Here, we propose SurvCNN, an alternative approach to process multi-omics data
    with robust computer vision architectures and predict cancer prognosis for Lung Adenocarcinoma patients.” But there is not image in this study, why computer vision?
  • Authors made two tables 1 and 2. In Table 1, they present twelve combinations of omics data analyzed in the study. Then in Table 2, they showed the overall statistics for all the omics data-types combinations for SurvCNN. Why this much obfuscations? Please remove table 1 and gather all the 6 combinations of I-VI by their names. Some readers don’t want to go for scavenger hunt inside the paper.

Reviewer 2 Report

The paper presents a machine learning framework for estimating cancer survival using multi-omics data in Lung Adenocarcinoma patients. The proposed technique transforms the multi-omics data to the images by mapping features to grayscale intensities and then feeds the images as the inputs to a convolutional neural network.  The paper is interesting application of ML. However, some essential information about the techniques is missed as follows:

  • The multi-omics data are transferred to 2 dimensional images using Distributed Stochastic Neighbor Embedding (t-SNE) according to section 2.3. There is no reference for this technique in this section (line 131), also there is not adequate information about how the transformation is performed or it is not clear in the text. Readers are interested to know how the multi-omics data are transferred to 2D space or how the features are mapped to grayscale intensities. The process could be explained in different steps or in form of an algorithm.
  •  Similarly, the CNN architecture is explained very briefly, missing the important details, for example, what is the size of training and testing datasets, what is the size of the input images, what is the output, etc.     

Reviewer 3 Report

An innovative CNN approach for survival analysis to integrate multiple data platforms from the TCGA (LUAD) is presented. The method is based on generating complex features from a numeric to image transformation (Sharma et al.). Further the authors evaluate various performance metrics and segregate patients into high-risk and low risk to e.g., to estimate differentially expressed genes. It is shown that integration of data platforms improves the performance. In particular the miRNA data with low number of 
features shows to not be suitable for the transformation, while the integration
of e.g., mRNA RNAseq and methylation data shows a prominent improvement over the single data platform. It is also shown that the method outperforms conventional appraches for LUAD.

-- I see one sentence selecting genes of the first and third quartile of the averaged pixel intensities compared between high and low risk to select differential "expressed" genes (denoted in the text as up and down regulated). I guess you mean first and fourth quartile right? How many genes are excluded and remain for this kind of selection? (Figure 3). Also unclear what the different individual omics are for these selected genes.

-- It still would be interesting to add a section of the relevance of actual gene locations and to which degree they improve the performance. Such as if (simulated) random UMAPs/tsne are likely to have a similar performance. Also defined reference structures maybe interesting to investigate if they perform well/ as good.

-- the actual omics integration is the averaging of grayscale pixels? Would it not be likely more usefull for the later interpretation to distinguish omics and have a larger feature space? For example in a naive case methylation and gene expression could be complementary (non-expressed gene with high methylation) so as to cancel each other in the gray scale? Would we not loose this information rather than having "unmerged" omics features, such as Gene1-mRNA, Gene1-met as separate features? Can you at least add this to the discussion section? Can you explain in more detail how features are "merged" across omics and if I understood this correctly?

-- Section 2.10
Does "KEGG 2019" denote access date? Also for "GO Biological processes (2018)"

-- show number of features before processing for LUAD (e.g. Table 2 or somewhere else separate), would also put in caption that this is LUAD dataset from TCGA, 

-- As the image transformation plays a major role and it would be
beneficial to explain the tabular to image data transformation steps with more detail. E.g. "various transformations". Also how the gene-locations
are mapped between omics. So you do not make separate t-sne/UMAPs for each omic separately (and you do it multi-dimensional?)?

-- It is one weak point of the study that only one omics data set was used

to evaluate e.g. the performance between methods.

Table 1.
why VI is not IX, order 1 dataset, 2-combo datasets, 3-combo datasets

Reviewer 4 Report

SurvCNN: Redefining the Utility of Multi-Omics Data in Cancer Survival Analysis

The authors developed a multi-omics machine learning predictive model, SurvCNN, to stratify patients with lung adenocarcinoma into low- and high-risk categories, which shows a significant concordance with patients’ overall survival. While the performance of SurvCNN appears to be better overall compared to other models, it is unclear how feasible and translatable SurvCNN is in the clinical setting, considering the requirement for multiple omics analyses from tissue biopsies. Also, the term “multi-omics” may be a bit misleading here since only the mRNA and DNA methylation data are contributing to the high-performance models, whereas the inclusion of miRNA produced poorer performance. While the paper is well-written overall, a reorganization of the data presentation (figure sequence) will greatly improve the flow of the paper.

Minor revision suggested:

  • Line 13, is “log-risk” supposed to be “low-risk”?
  • It would be helpful to provide additional information on how to interpret the performance metrics data in order to better appreciate the performance of the model.
  • Line 186, Figures S3-S8 are presented before S2. To make it easier to follow, figures and tables should be presented in a sequential manner. 
  • Figure 3C legend is missing. 
  • Figure 6B, more descriptive GO biological processes would make it easier to follow than the GO IDs.
  • Overall, the methods are difficult to follow and deficient in details, making it difficult to replicate the study.

Round 2

Reviewer 1 Report

Authors responded well to my questions. I have no further comments.

Reviewer 3 Report

I have read the revision and the manuscript substantially improved already. The paper is very valuable and all comments have been addressed satisfactorily.   Here are only minor comments/ revisions that I think are optional but would be helpful to the reader.   1) I had no access to the additinal file where the description of the 1D --> 2D feature transformation is said to be described in more detail. I wonder why this could at least very briefly added to the method section or at least a citation of the process. I see as mentioned in citation [31] a similar approach is described and it is difficult to assess if a different approach is presented or what the modifications are.   2) The roman numerals notation may be convenient but they make it difficult to interpret while reading what combination or individual data they are. E.g. instead of roman numerals a readable character code or something intuitive. MR, ME, MI, CL, ... MR-ME, MR-MI, ..., MR-ME-CL etc ... It would not be substantially longer strings compared to a roman numeral. However, this comment is optional.   3) The major weakness of the manuscript is still that only a single omics collection dataset LUAD from TCGA was used and or no simulated data. So the presented analysis is provided as a prove of concept. Maybe if not already mentioned in the text this statement could be added.   I understand that the results and features/genes from eg two different cohorts may not be concordant but the key observations e.g. on the pathway level and for example limitations of the feature dimensions and the comparisons to other methods would be expected to be comparable. However, I understand that this is difficult also due to runtime and the extensive curative efforts.